

# Technical Note: Improved Sampling of Behavioral Subsurface Flow Model Parameters Using Active Subspaces

Daniel Erdal[1] and Olaf A. Cirpka[1]

[1]University of Tübingen, Hölderlinstr. 12, 72074 Tübingen, Germany

**Correspondence:** Daniel Erdal (daniel.erdal@uni-tuebingen.de)

**Abstract.** In global sensitivity analysis and ensemble-based model calibration it is essential to create a large enough sample of model simulations with different parameters, which all yield plausible model results. This can be difficult if a-priori plausible parameter combinations frequently yield non-behavioral model results. In a previous study (Erdal and Cirpka, 2019), we developed and tested a parameter-sampling scheme based on active subspace decomposition. While in principle this scheme

worked well, it still implied testing a substantial fraction of parameter combinations that ultimately had to be discarded because of implausible model results. This technical note presents an improved sampling scheme and illustrates its simplicity and efficiency by a small test case. The new sampling scheme can be tuned to either outperform the original implementation by improving the sampling efficiency while maintaining the accuracy of the result, or by improving the accuracy of the result while maintaining the sampling efficiency.

## 1  Introduction

Global sensitivity analysis (e.g., Saltelli et al., 2004, 2008) is an established technique for quantifying the importance of uncertain parameters of a model. It has also gained popularity within hydrological sciences, with many different methods to choose from (e.g., Mishra et al., 2009; Song et al., 2015; Pianosi et al., 2016). An increasingly popular global-sensitivity approach is the method of active subspaces (e.g. Constantine et al., 2014; Constantine and Diaz, 2017). While been designed for

engineering applications (e.g. Constantine et al., 2015a, b; Hu et al., 2016; Glaws et al., 2017; Constantine and Doostan, 2017; Hu et al., 2017; Grey and Constantine, 2018; Li et al., 2019), it has recently been used with good performance in hydrology (e.g., Gilbert et al., 2016; Jefferson et al., 2015, 2017; Teixeira Parente et al., 2019), including a recent study of ourselves (Erdal and Cirpka, 2019).

A key issue when conducting a global sensitivity analysis, is the requirement of a large enough sample of model simulations

with parameters ranging over the full parameter space. Simulations showing unrealistic behavior (e.g., wells or rivers running dry in the model, while they in reality always have water) should be removed from the sample. Already in moderately complex models this may result in many model trials that must be discarded on the level of a plausibility check. This leads to the contradictory requirements of sampling the entire space of parameters defined by preset wide margins to capture the entire distribution while exploring only the part of the parameter space yielding plausible results. One way of easing the computational

burden, is to make use of a simpler model (i.e. surrogate/proxy/emulator model), discussed, e.g., in the comprehensive reviews





of Ratto et al. (2012), Razavi et al. (2012), Asher et al. (2015), and Rajabi (2019). A common sampling approach is to use a two-stage acceptance sampling scheme, in which a candidate parameter set is first tested with the surrogate model, and only if the surrogate model predicts the parameter set to be behavioral, it is applied in the full model. This idea has been applied to groundwater modelling by Cui et al. (2011), Laloy et al. (2013), and the authors of the current study (Erdal and Cirpka, 2019).

In the latter study, we used a response surface fitted to the first two active subspaces as the surrogate model in a sampling scheme for a subsurface catchment-scale flow model. The scope of the current technical note is to present an improvement of this scheme and compare it to the original one.

## 2 Methods

In the following subsections we briefly describe the active-subspace method and the base flow model. More details are given

by Erdal and Cirpka (2019).

### 2.1 Active Subspaces

In this section we briefly repeat the basic derivation of active subspaces for a generic function $f(\tilde{\mathbf{x}})$, in which $\tilde{\mathbf{x}}$ is the vector of scaled parameters $\mathbf{x}$ with a scaling to the range between 0 and 1. An active subspace is defined by the eigenvectors of the following matrix $\mathbf{C}$, computed from the partial derivatives of $f$ with respect to $\tilde{x}_i$, evaluated over the entire parameter space

(Constantine et al., 2014), here shown with its eigen-decomposition and Monte Carlo approximation (Constantine et al., 2016; Constantine and Diaz, 2017):

$$\mathbf{W}\mathbf{\Lambda}\mathbf{W}^{-1} = \mathbf{C} = \int \nabla f(\tilde{\mathbf{x}}) \otimes \nabla f(\tilde{\mathbf{x}}) \rho(\tilde{\mathbf{x}}) d\tilde{\mathbf{x}} \approx \frac{1}{M} \sum_{i=1}^{M} \nabla f(\tilde{\mathbf{x}}_i) \otimes \nabla f(\tilde{\mathbf{x}}_i) \tag{1}$$

in which $\otimes$ denotes the matrix product, $\rho$ is a probability density function, the integration is performed over the entire parameter space, $\mathbf{W}$ is the matrix of eigenvectors, $\mathbf{\Lambda}$ is the diagonal matrix of the corresponding eigenvalues, and $M$ is the number

of samples used. The $n$-dimensional active subspace is spanned by the eigenvectors with the $n$ highest eigenvalues. In our application, we use $n = 2$ as we could detect very little improvement with higher numbers.

In a global sensitivity analysis using active subspaces, the activity score $a_i$ of parameter $i$ is defined by:

$$a_i = \sum_{j=1}^{n} \lambda_j w_{i,j}^2. \tag{2}$$

in which $\lambda_j$ is the $j$-th eigenvalue and $w_{i,j}$ the element relating to parameter $i$ in the $j$-th eigenvector. In the following, we

consider the square root of the activity score to obtain a quantity that has the same unit as the target variable $f$.





## 2.2 Model Application

In our application we consider a model of the small Käsbach catchment in south-west Germany. The model has 32 unknown
parameters, including material properties, boundary-condition values, and geometrical parameters of subsurface zones. Origi-
nally, Erdal and Cirpka (2019) simulated subsurface flow in the domain using the model-software HydroGeoSphere (Aquanty
Inc., 2015), which solves the 3-D Richards-equation, here using the Mualem-van-Genuchten (Van Genuchten, 1980) parame-
terization for unsaturated flow. Figure 1 illustrates the model domain. Details, including the governing equations, are given by
in Erdal and Cirpka (2019).

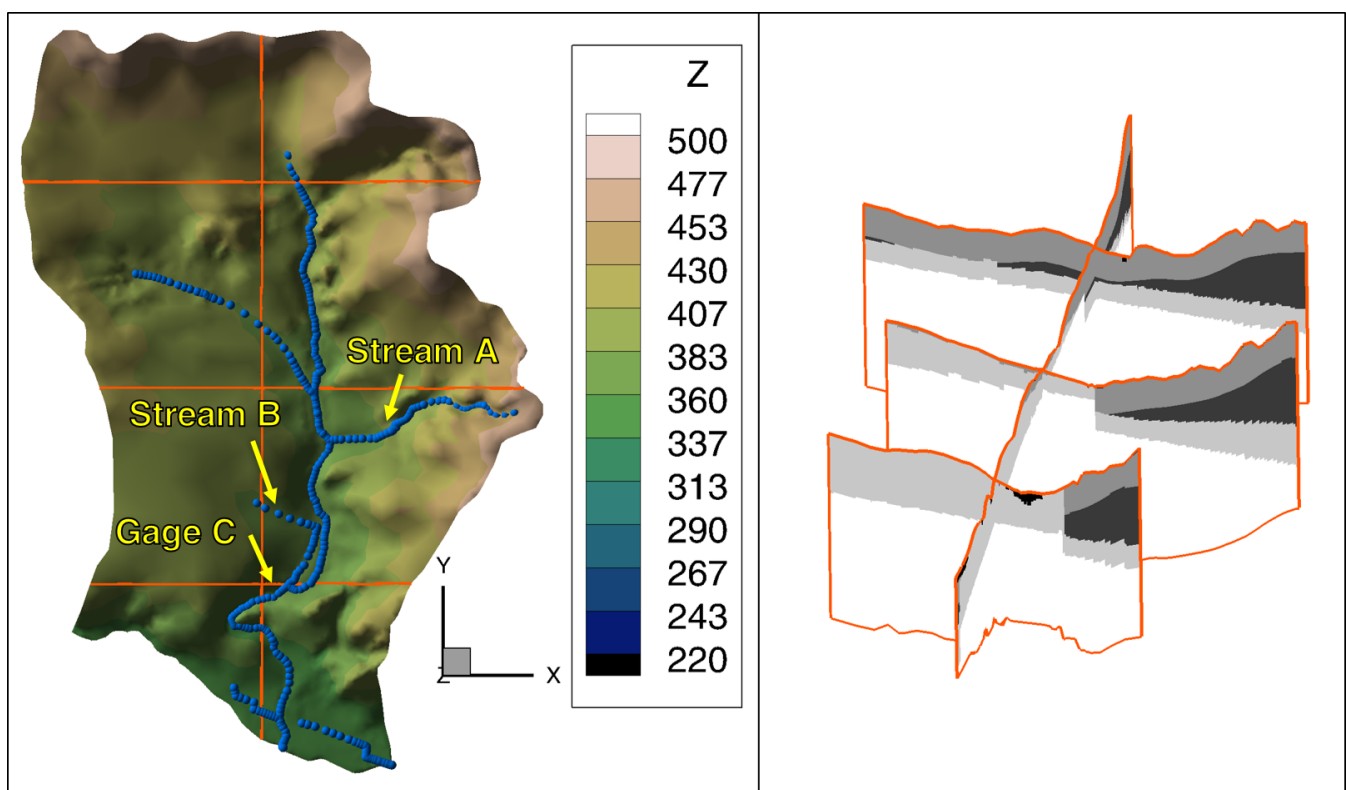

**Figure 1.** Illustration of the model domain. Left: shape of the domain and topography; right: example of a geological realization.

In a related study, we constructed a surrogate model using Gaussian Process Emulation (GPE) from roughly 4,000 parameter
sets. In the GPE model, the model response $f(\tilde{\mathbf{x}}_i)$ at the scaled parameter location $\mathbf{x}_i$ is constructed by interpolation from the
existing set of parameter realizations using kriging in parameter space with optimized statistical parameters. The GPE-model
is constructed with the Small Toolbox for Kriging (Bect et al., 2017). In the present work, we use the GPE-model instead of
the full HydroGeoSphere flow model as our virtually true model response. The prime reason for this is that we can perform
pure Monte Carlo sampling of behavioral parameter sets with the GPE model, requiring about 600,000 model evaluations to
create a set of 3,000 behavioral parameter-sets, which would be unfeasible with the original HydroGeoSphere model. That





is, we use a surrogate model (the GPE model) to judge the performance of other surrogate models (based on active-subspace decomposition) in creating ensembles of plausible parameter sets.

Like in our prior work (Erdal and Cirpka, 2019), the model considers 6 observations defining a behavioral performance (for locations see Figure 1):

- Limited Flooding: maximum of $2 \times 10^{-3}$ m$^3$/s of water leaving the domain on the top but outside of the streams

– Division of water: between 25-60% of incoming recharge leaves the domain via the streams.

- Gage C: minimum flow of $5 \times 10^{-3}$ m$^3$/s.

- Stream A: maximum flow $3 \times 10^{-3}$ m$^3$/s

- Stream B: minimum flow $5 \times 10^{-6}$ m$^3$/s.

With the aim of keeping this technical note rather concise, we will not discuss individual parameters or their meaning in the
model. To this end, we address all parameters by a parameter index (1-32) instead of a name, and the resulting histograms refer to the the scaled parameters, ranging from 0 to 1.

### 2.3   Sampling Schemes using Active-Subspace Decomposition

The basic idea of using a surrogate-assisted sampling scheme is to use the (very fast) surrogate model to first evaluate a candidate parameter-set. If the surrogate model predicts the parameter set to be behavioral, it is stage-1 accepted and will be
ran with the full model. If accepted also after running the full model, a parameter-set is stage-2 accepted. Only the stage-2 accepted parameter sets are used in the global sensitivity analysis, whereas the stage-1 accepted ones are used to improve the surrogate model.

For each observation considered, we need to perform an active-subspace decomposition. In our our previous work (Erdal and Cirpka, 2019), a decision on whether to accept or reject a parameter set is made in the following way:

1. A third-order polynomial surface is fitted in the active subspace spanned by the two major active variables.

     2. These polynomial surfaces are used to predict the observations of a candidate parameter-set.

     3a If all predicted observations are acceptable, the candidate is stage-1 accepted.

     3b If any predicted observation is between the acceptance point and a user-defined outer point, we assign a probability of being stage-1 accepted by linear interpolation between 0 (at the outer point) and 1 (at the acceptance point), draw a
random number from a uniform distribution, and stage-1 accept the parameter set if the assigned probability is larger than the random number.

     3c If any predicted observation is outside of the outer point, we reject the sample, draw a new candidate, and return to (2).




4 After adding 100 stage-1 accepted parameter sets, we recalculate the active subspace using all stage-1 accepted parameter sets collected to this point.

Two critical points can be seen with this scheme. First, the polynomial surface is fitted through all stage-1 accepted points across the entire parameter space. However, locally, where we wish to make a prediction, it could still be strongly biased. Second, the user needs to prescribe the outer-points, which should not only cover our uncertainty about the acceptance point, but also implicitly addresses the error by using the active-subspace decomposition. As we project 32 dimensions to two, the potential for an imperfect decomposition is rather high (that is, two close points in active subspace may have different

behavioral status). As we have no rigorous and yet simple method to address this uncertainty, the choice of the outer point becomes fairly subjective.

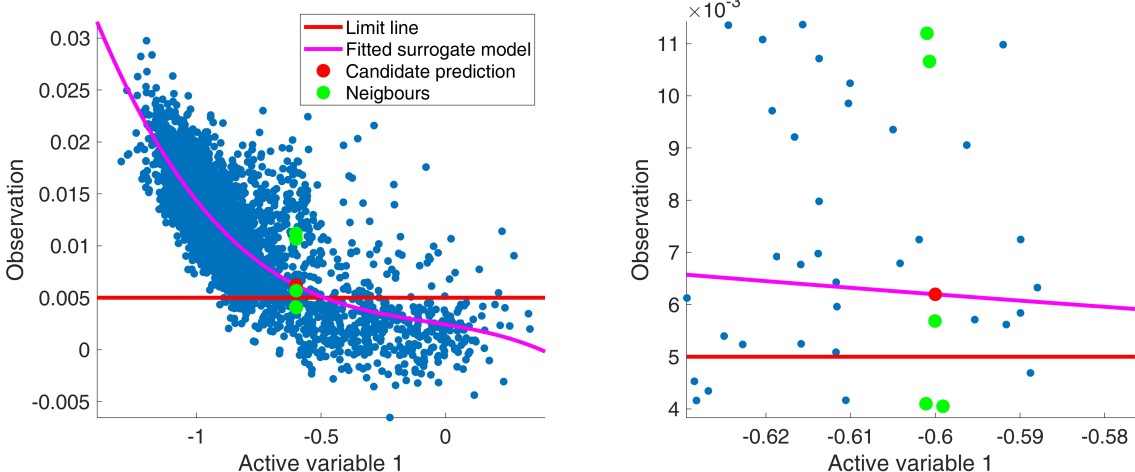

**Figure 2.** Illustration of the two active-subspace sampling schemes, shown for a 1-D test. The right plot shows a zoom-in into the left plot. Blue dots: previously analyzed points; magenta line: fitted polynomial surrogate model; red dot: candidate parameter in active subspace (x-value) with the assigned polynomial prediction (y-value) of the original sampling scheme; green dots: neighbors considered in the new scheme, which are chosen exclusively by the active-variable value; red line: acceptance criterion.

    To overcome these these issues, we here suggest a modified sampling scheme, with fewer tuning parameters and less sensitivity to local biases. As with the original scheme, we require one active subspace decomposition per observation and use the first two active variables to create the two-dimensional active subspace. The process is as follows:

1. The candidate parameter set is projected into the active subspace.

     2. The closest neighbors in the active subspace are sought. In this work we use the 5 closest neighbors plus all neighbors that fall within an ellipse around the candidate point that has a radius of 1% of the total range of each active subspace, in each of the two dimensions.





3. For each observation, a candidate parameter-set is pre-accepted if a certain ratio ($P$) of its neighbours are behavioral (i.e., stage-2 accepted).

4. The candidate parameter-set is stage-1 accepted if it was pre-accepted for all observations, otherwise it is rejected.

5. If rejected, draw a new candidate parameter set and return to (1).

Like before, we recalculate the active subspace after adding 100 stage-1 accepted parameter sets. The two approaches are illustrated in Figure 2, although just for a 1-D illustrative example. As can be seen in the figure, the original sampling scheme suggests that the candidate is behavioral (red dot is above the red line). With the new sampling scheme, on the other hand, it becomes a matter of the $P$-value chosen. At $P = 0.15$ and $P = 0.55$, the candidate would have been stage-1 accepted (60% of the green dots are behavioral), while at $P = 0.75$ the candidate would have been rejected. In this work, we consider the ratios $P = 0.15$, $P = 0.55$ and $P = 0.75$, and compare the performance of the sampling scheme with that used in the previous study (Erdal and Cirpka, 2019).

## 3 Results and Discussion

Figure 3 shows the acceptance ratios for the original sampling scheme and the new sampling scheme with three different $P$-values, together with a pure Monte-Carlo sampler without preselection, applied to the Käsbach GPE-model with 32 parameters. As can be seen, the new scheme with $P = 0.75$ is the fastest, while the original scheme and the new scheme with $P = 0.15$ show rather comparable behavior with lower acceptance rates. For comparison, the pure Monte Carlo sampling has an acceptance ratio of $\approx 0.5\%$.

While high acceptance rates are favorable in light of computational efficiency, we also want to avoid introducing a bias by the preselection scheme. We evaluate such bias, by considering the marginal parameter distributions of the stage-2 accepted samples, which should agree with the distribution obtained by the (inefficient) pure Monte-Carlo sampler. Figure 4 shows the resulting histograms for the three parameters with the most complex marginal distributions. We quantified the agreement of the marginal distributions of the sampling schemes with preselection and the pure Monte-Carlo sampling by the Cramér–von Mises metric $\omega^2$:

$$\omega^2 = \int\limits_0^1 \left( \hat{P}_{ss}(\tilde{x}_i) - \hat{P}_{MC}(\tilde{x}_i) \right)^2 d\tilde{x}_i \tag{3}$$

in which $\hat{P}_{ss}(\tilde{x}_i)$ is the marginal cumulative probability of the scaled parameter $\tilde{x}_i$ for a tested sampling scheme and $\hat{P}_{MC}(\tilde{x}_i)$ is the same quantity for pure Monte-Carlo sampling. The corresponding values of $\omega^2$ are reported in the subplots of Figure 4.

From the histograms in Figure 4 and the values of the Cramér–von Mises metric $\omega^2$ it becomes obvious that the fast new sampling with $P = 0.75$ results in marginal distributions that significantly differ from those of the unbiased pure Monte-Carlo scheme. The new scheme with $P = 0.55$ results in marginal distributions that are comparable to those of the original scheme,




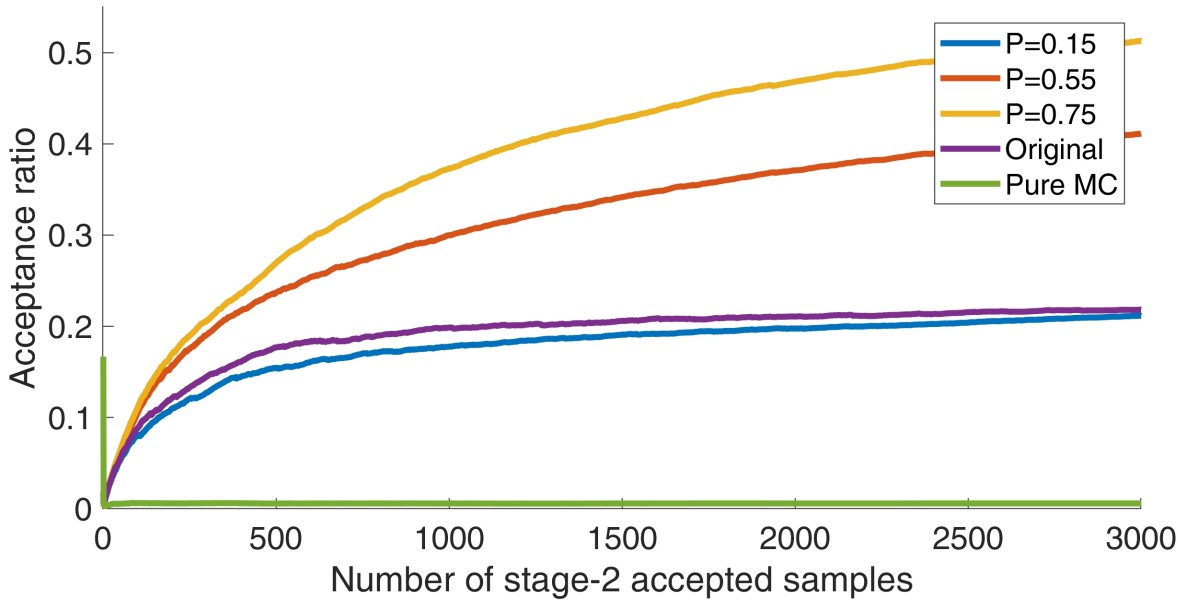

**Figure 3.** Acceptance ratios of the different sampling schemes, plotted as a function of the number of stage-2 accepted samples.

but that have been achieved by a sampling scheme with twice the acceptance rate and thus half the computational effort. By contrast, the new scheme with $P = 0.15$, which caused a computational effort similar to the original scheme, resulted in a

marginal posterior distribution that is very similar to that obtained by pure Monte-Carlo sampling. Hence, we can conclude that the proposed sampling scheme is superior to the old one: either it has much better sampling accuracy for the same efficiency ($P = 0.15$), or it has a much better efficiency with a very comparable accuracy ($P = 0.55$).

Figure 5 shows the square-root of the activity score for a selected target variable, computed by the active-subspace based global sensitivity analysis and using the different sampling schemes, which confirms the impression of the histograms shown

in Figure 4. The pure-MC scheme and the new scheme with $P = 0.15$ show almost identical activity scores, while the score-patterns increasingly differ with increasing $P$-values. Similarly, the original sampling scheme differed in the activity scores compared to the pure-MC scheme. Nonetheless, all sampling schemes correctly identified the two most important parameters and the correct set of the ten most important parameters. That the order of the parameters within the set of the most important parameters is not captured by the faster sampling schemes may be an acceptable trade-off between speed and accuracy, 150    depending on the individual application.

In the current study, we have used Gaussian process emulation (GPE) as a proxy of the full HydroGeoSphere model, putting the question forward whether a GPE model could not also be used as surrogate model for preselection in an advanced sampling scheme. This is indeed possible, and we are currently developing such schemes, achieving acceptance ratios between 70-90%. Hence, GPE-based sampling schemes can be notably more efficient than the new scheme presented in this work. Nonethe-

less, we see a clear value in using the less efficient active-subspace based sampling schemes. The key word is simplicity.





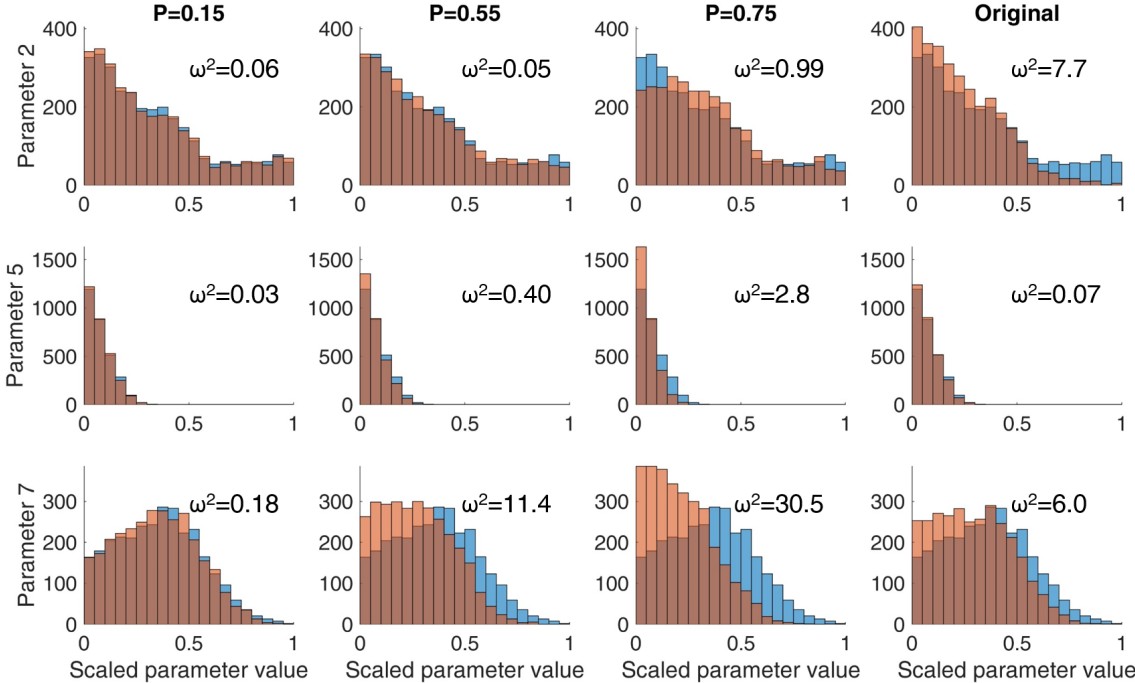

**Figure 4.** Histograms of the three parameters with the most complicated posterior marginal distributions. Each row shows a parameter and each column a sampling scheme. Blue bars: histograms from pure Monte Carlo sampling (i.e. true distribution); brown bars: sampling schemes with preselection; numbers: Cramér–von Mises metric $\omega^2$ for the distance between the two distributions, here shown multiplied with 1000 for increased readability.

The full active subspace-sampling scheme is implemented in-house, and the most complicated step is likely the eigenvalue decomposition, which is a standard tool in any programming environment. Hence, we have full control over the entire selection procedure. Further, the active-subspace based sampling scheme presented here has a single tuning coefficient $P$ with an easily comprehensible meaning, and the resulting active subspace can easily be visualized for an intuitive understanding of the method. This is quite different with GPE-based methods which require choosing a covariance function in parameter space with coefficients that needs to be estimated from the current set of training data. In our application, we have 32 original parameters, requiring one variance and 32 integral scales as covariance coefficients to be estimated every time the GPE-model is re-trained. Estimating 33 covariance parameters from $\mathcal{O}(1000)$ parameter sets is time consuming, and the integral scales in non-sensitive parameter directions are not well constrained by the data at all. Finally, to train a GPE model we need to rely on third-party codes which remain black boxes to a large extent, and usually involve a rather decent amount of work until they do what they are supposed to do. Hence, we clearly see a benefit of using the simpler active-subspace based sampling schemes even if they are computationally less efficient.

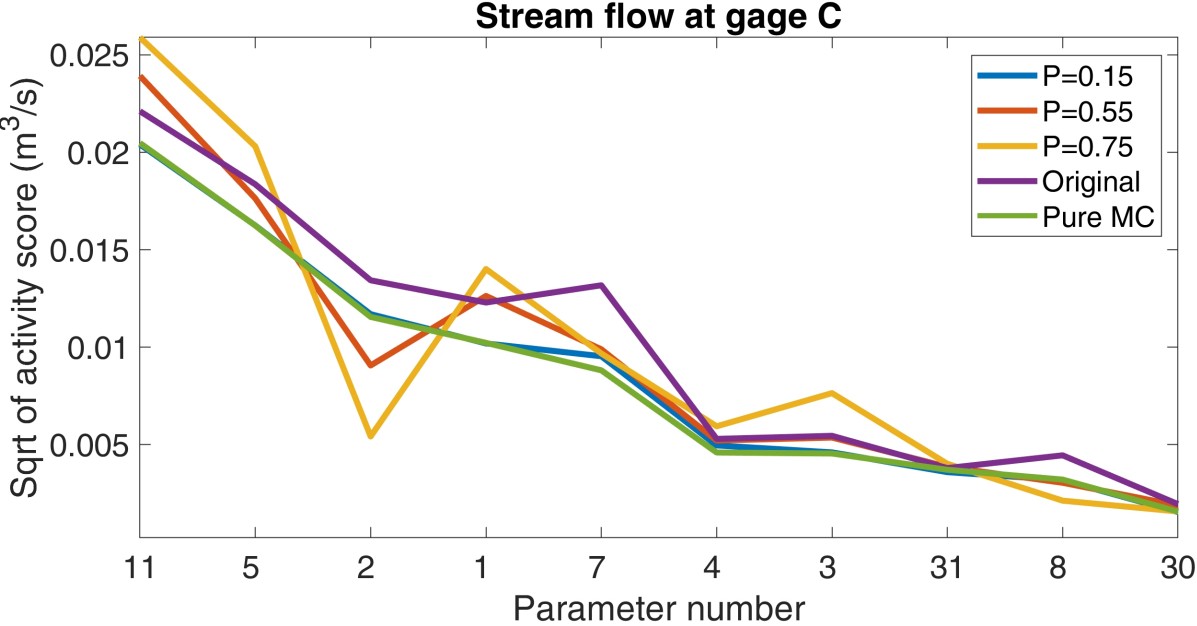

**Figure 5.** Square-root of activity scores of the 10 most influential parameters for the target variable stream flow at gage C resulting from applying the active-subspace based global sensitivity analysis to the posterior distributions using the different sampling schemes.

## 4    Conclusions

In this work we have presented an improved sampling scheme to obtain ensembles of parameter sets that lead to plausible model results. Like in the preceding study of Erdal and Cirpka (2019), the sampling scheme makes use of an active-subspace based preselection scheme that reduces the number of full model runs that need to be discarded. In contrast to the preceding method, we don't perform a polynomial fit over the entire parameter space anymore, neither do we have to set fuzzy boundaries of the target variables to define the behavioral status. Instead, the preselection of a parameter set is simply based on the behavior of surrounding trial solutions. The new scheme outperforms the preceding one by either achieving a higher accuracy in the resulting posterior parameter distributions for the same sampling efficiency, or by having a much higher sampling efficiency for a comparable accuracy. We hence conclude that the new scheme presented here should be used instead of the original one.

*Code availability.* All own-developed codes necessary to run the Stochastic Engine used in this work are in the process of being transferred to a permanent repository at the university of Tuebingen. This will be completed before the paper is revised. In the mean while, exactly the same codes are available from https://github.com/d-erdal/StochasticEngine





*Author contributions.*   Simulations and code development were performed by DE. Both authors contributed to developing and writing the paper. OAC was responsible for acquisition of the funding.

*Competing interests.*   No competing interests

*Acknowledgements.*   This work was supported by the Collaborative Research Center 1253 CAMPOS (Project 7: Stochastic Modeling Framework of Catchment-Scale Reactive Transport), funded by the German Research Foundation (DFG, Grant Agreement SFB 1253/1).





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
