# Peer review of "Technical Note: Improved Sampling of Behavioral Subsurface Flow Model Parameters Using Active Subspaces"

_Hydrology and Earth System Sciences, 2019_

## Referee Comment (RC1) · Anonymous Referee #1 · 10 Mar 2020

Content Comments:

1. Page 5, Figure 2 - Consider labeling the red line "Behavioral Limit Line" for clarity. Can one assume the point has to be above the limit line to be considered acceptable behavior? Could Figure 2 be moved so that it is after Line 115?

2. Page 5, Line 105 – Where does the active subspace come from that the initial candidate parameter sets (say, the first 1-99) are projected onto? Line 113 states that the active subspace is recalculated after adding 100 state-1 accepted parameter sets – but, how do you start?

3. Page 5, Line 106 – Can you provide any insight about how the values/criteria (e.g., 5

closest neighbors plus 1% radius) were selected for this work that would be beneficial for another researcher trying to implement this method?

4. Page 6, Line 121 – Is the "acceptance ratio" the ratio of candidates that are stage-1 accepted to the total number of candidate parameter sets (stage-1 accepted + rejected)? Or, is the "acceptance ratio" the ratio of candidates that are stage-1 accepted to those that are stage-2 accepted (i.e., the amount of pre-accepted candidates that become accepted). This clarification would also help interpret Figure 3.

5. Page 6, Line 121 and 136 – Intuitively, I am struggling to understand why P=0.75 is the fastest when it should, in my mind, be the most difficult to achieve. And, along those lines, why P=0.75 sampling results in a significantly different distribution from the unbiased pure Monte-Carlo scheme. Do you have any insight into why this is occurring?

a.    Furthermore, do you think the P value selected is dependent on the model/application? Based on your experience, is the exercise of comparing different P values and selecting one necessary for another researcher trying to implement this method, or do you think the P=0.55 scheme is broadly applicable?

Grammar Comments:

1. Page 4, Line 67 – Line states that the model considers 6 observations, but there are only 5 listed below this sentence. Should 6 be changed to 5?

2.  Page 4, Line 67 – Consider revising the sentence to state "...observations that define acceptable behavioral performance..."

3. Page 4, Lines 69-73 – Make the list style consistent in regard to the period placement at the end of each list item (or remove them all).

4. Page 4-5, Lines 87-93 – Add period after list item number 4.

5. Page 6, Lines 109 and 111 – Remove hyphen between "parameter-set".

6. Page 6, Line 125 – Present the acceptance ratio at 0.005 (not a percent) since the acceptance ratios are shown as decimal values on the y-axis of Figure 3.

---

## Referee Comment (RC2) · Anonymous Referee #2 · 12 May 2020

The current work presents a sampling strategy for those cases when some values of the investigated model output(s) are classified as unfeasible/unacceptable and the corresponding parameters sets are labelled as non-behavioral.

A key step is the transformation from the original N-parameters space into the space spanned by the n-most relevant eigenvectors (here two are considered, i.e., n = 2). Then, in this reduced dimensional space an active region is identified (i.e., the active subspace) is identified (see Fig. 2 where all the space above the red line is the active subspace). Then a set of parameters is chosen to be behavioral or not (i.e., the associated output(s) belongs to the active subspace o not) in a two stages approach:

(1) a surrogate model of dimension n in the space spanned by the n-most influential eigenvector is built and then used to check if a parameter set is behavioral or not, as a 'first approximation'; (2) if a parameters set passes stage-1 the full model is run for that parameters set a second check on being or not behavioral is done. Then only stage-2 parameter sets are retained for successive analysis. The main gains here are due to the reduction of the dimension (from N to n) and the use of a surrogate model in the n-dimensional space to skim those parameters sets that are not behavioral.

The improvement/modifications proposed in the current work are during stage-1, where an additional constrain is added: a parameters set passes stage-1, if in its neighbor-hood there is a certain fraction P of parameter sets that have already passed stage-2.

The paper is of interest and well written. There are some unclear (at least to me) points which I would like to be addressed before publication, hoping for a more clear and more accessible work after revision.

Comment 1 In both approaches, after 100 parameters sets passed stage-1 the eigen-vector decomposition is re-done, and so the surrogate in the n-space dimensions is built again. My understanding is that the output(s) values associated with these 100 samples are obtained through the n-dimensional surrogate model (before adding the 100 samples), right? If this is the case, isn't there the risk of 'guiding'/'move'/'bias' the active subspace toward the results of the surrogate model? For example, in Fig. 2 the new extra 100 stage-1 accepted points will all falls along the purple curve (along its branch above the red line). This could be an issue if the surrogate is doing a poor job. Am I wrong?

Why not use stage-2 accepted sample (even though they require full model runs) to update the eigenvectors/eigenvalues? This will avoid the issues associated with a possibly poor surrogate modelling.

Comment 2 How is the algorithm initialized? Which is the size of the sample to build the first n-dimensional subspace? How is relevant? For example, in Fig. 2 there are

previously analyzed parameters samples/output, they should come from a set of full model runs (then they are updated after 100-samples pass stage-1t, see the previous comment).

Comment 3 Acceptance ratio: this the ratio between the stage-2 accepted sample and the drawn samples, right?

Why is it a function of the stage-2 accepted samples (see Fig. 3)? I don't see this aspect being used in the algorithm (both previous and current versions) at any step. I would have expected a dependence on the stage-1 accepted samples.

Moreover, as P (i.e., the fractions of neighborhood accepted, at least at stage-1, samples) increases I would expect lower acceptance ratios, i.e., it becomes harder for a sample set to be accepted as a larger fraction of its neighbors have to be in the active subspace (i.e., P increases). (see also lines 116-117 that go along this line of reasoning). Please clarify.

Isn't that, since P is the exact fraction (not an exceedance fraction) of good neighbors, as P increases the active subspace is updated (on top of 100 samples that pass stage-1) by favoring those regions of the active subspace that are the most distant from the threshold condition (e.g., upper left part of Fig. 3a) where it is more easy to have P high than low? Then, the n-dimensional surrogate will be update by favoring these far-from threshold condition regions leading to a poor behavior (due to its global character) in those regions close to the threshold conditions (e.g., lower right region in Fig. 3). This is then reflected in the decreased quality of the behavioral parameters pdfs as shown in Fig. 4. Or maybe, I am just speculating too much here. It could be of interest to see how the n-dimensional surrogates evolve as a function of P, for example after some updates are conducted to see if there is this tendency or not.

Comment 5 Since a surrogate model is used to mimic also the full model response (see Sec. 2.2) I would suggest to refer to this as 'full-model-surrogate' in order to mark the distinction with the surrogate model build in the n-dimensional space.

---

## Author Response (AR1)

**Final Response to the Reviewer Comments**

**Technical Note: Improved Sampling of Behavioral Subsurface Flow Model Parameters Using Active Subspaces**

manuscript hess-2019-629

18.06.2020

Daniel Erdal & Olaf A. Cirpka

We would like to thank the editor and the two anonymous reviewers, whose constructive comments helped improve the manuscript. In the following pages, we provide detailed answers to each comment. In summary, the following three main changes are applied to the manuscript:

1. We define the term "acceptance ratio" better.

2. We describe the initial sampling.

3. We better explain the non-intuitive result concerning the higher $P$-value cases.

To aid the reading, the original comments by the reviewers are displayed in black, while our replies are both indented and blue. Line numbers in the reviewers comments refers to the original submission, while line numbers in our replies refers to the revised manuscript.

**Editor**

The authors present a sampling method to select parameters leading to plausible model results. The manuscript has been revised by 2 independent reviewers. All reviewers consider the study of (potential) interest for HESS after moderate revisions. All reviewers provide detailed comments and suggestions on diverse aspects of the manuscript. One aspect that the authors should consider during the revision of their work is to further elucidate the metric used for the global sensitivity analysis performed, GSA. Although Sensitivity is an intuitive concept, a variety of approaches/metrics have been proposed in the literature. Since each metric focuses on a different property of the model response(s), diverse metrics may lead to different results (Razavi and Gupta 2016, Water Resources Research, 51, 5, doi: 10.1002/2014WR016527; Dell'Oca et al. 2017, Hydrology and Earth System Science, 21,12, doi: 10.5194/hess-21-6219-2017).

We would like to thank the editor for taking time to handle our manuscript. As for our choice of GSA-metric, the editor is right that we did not discuss this point in the manuscript. The suggested articles are highly relevant for this purpose, and we have added them together with a brief discussion on the definition of the used GSA-metric. However, the main purpose of this manuscript is the development of the selection method within the sampling scheme, and in the context of the paper the GSA-analysis itself is merely one way of comparing the results of the sampling. The following text has been added (lines 50-57):

"It should be noted that there are different global sensitivity methods with different metrics that may give different results (e.g. Razavi and Gupta, 2015; Dell'Oca et al., 2017). In principle, nothing speaks against computing another global-sensitivity metric for the sample selected by our active-subspace based sampling scheme, as long as computing the metric is based on a random sample. For practical reasons and for a direct comparison with our previous work, we use the activity score in the present study. For the interested reader, a longer discussion about the current metric in relation to the specific application is given by Erdal and Cirpka (2019), and more general discussions have been presented by Saltelli et al. (2008); Song et al. (2015); Pianosi et al. (2016), among others."

**Anonymous Referee #1**

We thank the reviewer for taking the time to review our manuscript and providing constructive comments. In the following we address all individual comments one-by-one.

**Content Comments**

1. Page 5, Figure 2 - Consider labeling the red line "Behavioral Limit Line" for clarity. Can one assume the point has to be above the limit line to be considered acceptable behavior? Could Figure 2 be moved so that it is after Line 115?

We use the improved label suggested by the reviewer. The figure is also moved as suggested, but the final typesetting is not in our control. The reviewer is correct: a point has to be above the limit line to be considered acceptable. This is also further highlighted in the figure caption. Figure with new caption is found between lines 136-137 on page 6 in the revised manuscript.

2. Page 5, Line 105 – Where does the active subspace come from that the initial candidate parameter sets (say, the first 1-99) are projected onto? Line 113 states that the active subspace is recalculated after adding 100 state-1 accepted parameter sets– but, how do you start?

We thank the reviewer for highlighting this point, as the information is clearly missing. The initial active subspace is created from a random set of 50 parameter sets drawn from the unconditional prior using Latin Hypercube sampling. The following text is added to the manuscript (lines 115-117):
"As in the original sampling scheme, we start with a set of 50 candidate parameter sets, sampled using a Latin Hypercube setup, which are per definition directly stage-1 accepted."

3. Page 5, Line 106 – Can you provide any insight about how the values/criteria (e.g., 5 closest neighbors plus 1% radius) were selected for this work that would be beneficial for another researcher trying to implement this method?

As the reviewer correctly points out, this is an important aspect of the sampling scheme. In our case, the values were chosen based on prior tests. However, the results were not highly dependent on the choice. The 1% was chosen just to ensure that the 5 neighbors are not so close to the candidate point that relevant uncertainties are neglected. We would consider both values to be applicable also to other model setups.
In the text, the following is added (lines 123-125): "The number of neighbors selected and the radius of the ellipse are tuning parameters, here chosen based on a few prior tests. However, we believe they are applicable also for other applications, at the very least as good starting points."

4. Page 6, Line 121 – Is the "acceptance ratio" the ratio of candidates that are stage-1 accepted to the total number of candidate parameter sets (stage-1 accepted + rejected)? Or, is the "acceptance ratio" the ratio of candidates that are stage-1 accepted to those that are stage-2 accepted (i.e., the amount of pre-accepted candidates that become accepted). This clarification would also help interpret Figure 3.

A very relevant comment that was not clearly explained. Alternative number 2 is what we meant (the ratio of candidates that are stage-1 accepted to those that are stage-2 accepted).
In the text, the following is added (new text within ") (line 138): "... shows the acceptance ratios '(number of stage-2 accepted samples divided by the number of stage-1 accepted samples)' or the original sampling scheme ..."

5. Page 6, Line 121 and 136 – Intuitively, I am struggling to understand why P=0.75 is the fastest when it should, in my mind, be the most difficult to achieve. And, along those lines, why P=0.75 sampling results in a significantly different distribution from the unbiased pure Monte-Carlo scheme. Do you have any insight into why this is occurring?

We understand the reviewers difficulty, indeed one would think that the sampling scheme which requires the highest number of correct neighbours would be most difficult and therefore most correct. This is also true when we look at stage-1 acceptances only. Here, P=.75 results in many more discharged candidate points, and of the stage-1 accepted ones, many are also stage-2 accepted (high ratio in Figure 3). The drawback, however, is that this sampling behavior effectively avoids sampling the boundaries between the behavioral and non-behavioral parameter space (it samples only "safe" parameter sets). This leads to a poor match when comparing to the pure Monte Carlo sampling (which samples everything and one uses stage-2 acceptance).

In the text, the following is added (lines 164-168): "While it may seem counter-intuitive that the highest P-values gets the highest acceptance ratio and the poorest match of the marginal distributions, it is worth noting that a higher P-value means that the requirement for stage-1 acceptance is higher. Hence, at high P-values we only sample the interior of the behavioral parameter space and avoid the boundaries where the behavioral status of a candidate parameter set is more uncertain. This results in the bias clearly seen in Figure 4."

5a. Furthermore, do you think the P value selected is dependent on the model/application? Based on your experience, is the exercise of comparing different P values and selecting one necessary for another researcher trying to implement this method, or do you think the P=0.55 scheme is broadly applicable?

In this work, =0.55 has been shown to be the best comprise between efficiency and accuracy, while also the P=0.15 case could be considered a good choice. We believe that either of these two, or a value in between is a generally applicably good for any sampling scheme, at least as a starting point.

The following text is added to the text (new text within ") (lines 176-177): "... captured by the faster sampling schemes may be an acceptable trade-off between speed and accuracy, depending on the individual application. 'Based on the experience gained within this project, a recommended starting P-value for our presented sampling scheme is P=0.55.'"

**Grammar Comments**

1. Page 4, Line 67 – Line states that the model considers 6 observations, but there are only 5 listed below this sentence. Should 6 be changed to 5?

2. Page 4, Line 67 – Consider revising the sentence to state "...observations that define acceptable behavioral performance..."

3. Page 4, Lines 69-73 – Make the list style consistent in regard to the period placement at the end of each list item (or remove them all).

4. Page 4-5, Lines 87-93 – Add period after list item number 4.

5. Page 6, Lines 109 and 111 – Remove hyphen between "parameter-set".

6. Page 6, Line 125 – Present the acceptance ratio at 0.005 (not a percent) since the acceptance ratios are shown as decimal values on the y-axis of Figure 3.

We thank the reviewer for carefully reading our manuscript and suggesting grammatical correction. All suggestions are followed in the revised manuscript.

**Anonymous Referee #2**

The current work presents a sampling strategy for those cases when some values of the investigated model output(s) are classified as unfeasible/unacceptable and the corresponding parameters sets are labelled as non-behavioral.A key step is the transformation from the original N-parameters space into the space spanned by the n-most relevant eigenvectors (here two are considered, i.e., n = 2).Then, in this reduced dimensional space an active region is identified (i.e., the active subspace) is identified (see Fig. 2 where all the space above the red line is the active subspace). Then a set of parameters is chosen to be behavioral or not (i.e., the associated output(s) belongs to the active subspace o not) in a two stages approach: (1) a surrogate model of dimension n in the space spanned by the n-most influential eigenvector is built and then used to check if a parameter set is behavioral or not, as a 'first approximation'; (2) if a parameters set passes stage-1 the full model is run for that parameters set a second check on being or not behavioral is done. Then only stage-2 parameter sets are retained for successive analysis. The main gains here are due to the reduction of the dimension (from N to n) and the use of a surrogate model in the n-dimensional space to skim those parameters sets that are not behavioral. The improvement/modifications proposed in the current work are during stage-1, where an additional constrain is added: a parameters set passes stage-1, if in its neighbor-hood there is a certain fraction P of parameter sets that have already passed stage-2. The paper is of interest and well written. There are some unclear (at least to me) points which I would like to be addressed before publication, hoping for a more clear and more accessible work after revision.

> We would like to thank the reviewer for her/his positive view on our work. All comments are addressed below.

Comment 1. In both approaches, after 100 parameters sets passed stage-1 the eigen-vector decomposition is re-done, and so the surrogate in the n-space dimensions is built again. My understanding is that the output(s) values associated with these 100 samples are obtained through the n-dimensional surrogate model (before adding the 100 samples), right? If this is the case, isn't there the risk of 'guiding'/'move'/'bias' the active subspace toward the results of the surrogate model? For example, in Fig. 2 the new extra 100 stage-1 accepted points will all falls along the purple curve (along its branch above the red line). This could be an issue if the surrogate is doing a poor job. Am I wrong? Why not use stage-2 accepted sample (even though they require full model runs) to update the eigenvectors/eigenvalues? This will avoid the issues associated with a possibly poor surrogate modelling.

> There seems to be a slight misunderstanding which may require a better explanation from our side. The reviewer is right that all surrogate-model samples will lie on the purple line in Fig.2 (or, in reality a surface in 2-D). However, this information is only used to compare against the user defined limit (red line in Fig.2) to decide if the parameter set is to be run in the full model or not. If the surrogate model is not doing a good job, the new points will be far away from the purple line. 100 of these full flow model runs are required before the update is performed. Hence, all samples used to train the active subspace and the surrogate model are full model runs. We think this might be part of the confusion and we stress this point much clearer in the revised manuscript (see changes applied below).
>
> Using just the stage-2 accepted samples would not be very beneficial for our purpose, which is to explore the full behavioral parameter space, since the surrogate model would only be good within the behavioral space, but rather poor at the boundary. This point, however, has well been discussed in our preceding HESS publication (Erdal & Cirpka, 2019), on which the present technical note is based. In order to keep the technical note brief, we avoid to discuss it here again.
>
> To increase the clarity of the manuscript we add the following:
>
> Description of surrogate model (lines 92-94): "Also, as the surrogate model is only used as a preselection filter, all results and the training of the surrogate model are based exclusively on full-flow model simulations." Sampling scheme point 4 (line 106): "Hence, the surrogate-model is based on all currently available full-flow model simulations. "

Comment 2. How is the algorithm initialized? Which is the size of the sample to build the first n-dimensional subspace? How is relevant? For example, in Fig. 2 there are previously analyzed parameters samples/output, they should come from a set of full model runs (then they are updated after 100-samples pass stage-1t, see the previous comment).

As also pointed out in comment 2 from reviewer 1, this information was clearly missing. The first 50 parameter sets, which are also the fist 50 full model runs, are sampled randomly from the unconditional prior using Latin Hypercube sampling. After the first 50 parameter sets are run, the first active subspace/surrogate model is built based on these runs, and then subsequently updated in 100 full model run intervals. In principle, we do not think that the method of initialization is that relevant, just as long as a reasonable coverage of the parameter space is achieved.

We have improved the manuscript by the following addition (lines 116-119): "As in the original sampling scheme, we start with a set of 50 candidate parameter sets, sampled using a Latin Hypercube setup, which are per definition directly stage-1 accepted. Hence we run the full flow model 50 times to initialize the sampling scheme. The actual number is not critical, and should be chosen with consideration to the number of unknown parameters."

Comment 3. Acceptance ratio: this the ratio between the stage-2 accepted sample and the drawn samples, right? Why is it a function of the stage-2 accepted samples (see Fig. 3)? I don't see this aspect being used in the algorithm (both previous and current versions) at any step. I would have expected a dependence on the stage-1 accepted samples. Moreover, as P (i.e., the fractions of neighborhood accepted, at least at stage-1, samples) increases I would expect lower acceptance ratios, i.e., it becomes harder for a sample set to be accepted as a larger fraction of its neighbors have to be in the active subspace (i.e., P increases). (see also lines 116-117 that go along this line of reasoning). Please clarify.

We see the reviewers confusion, as this was probably not explained in the manuscript (see also comment 4 from reviewer 1). The acceptance ratio is the ratio between the number of full model runs that are stage-2 accepted, and the total number of full model runs (which is the same as the number of stage-1 accepted model runs). In the manuscript we did not report about the total number of drawn parameters-sets, but this number is much (much!) higher than the number of stage-1 accepted samples. Further, the number of stage-1 rejected parameter-sets is by far the largest in the high P case. This results in a collection of stage-1 accepted samples that poorly explores the behavioral parameter space, but where a majority of the them are stage-2 accepted. Hence, the high-P case has a high acceptance ratio.

The difference between rejected and stage-1 accepted samples and their influence on the result has been clarified in the manuscript:

1) See answers to comments 4 and 5 from reviewer 1

2) Added in the introduction of the sampling scheme (lines 91-92): "Hence, one of the beauties of the surrogate-assisted sampling is its ability to quickly discharge large quantities of non-behavioral parameter-set without running the full flow model for each one (i.e. stage-1 rejected samples)."

and

3) Added to the results section (lines 142-147): "It should be noted here that the acceptance-ratio as a statistic only shows the ratio between the runs that are behavioral after running the full-flow model (stage-2 accepted) versus the number of full-flow model runs (stage-1 accepted). This, however, does not reflect the number of stage-1 rejected parameter sets, which is not reported in this work, but is by far the largest for the higher $P$-values. Hence, the acceptance-ratio is a measure of computational efficiency rather than a measure of search efficiency (which here is simple Monte Carlo and, hence, comparably inefficient)."

Comment 4. Isn't that, since P is the exact fraction (not an exceedance fraction) of good neighbors, as P increases the active subspace is updated (on top of 100 samples that pass stage-1) by favoring those regions of the active subspace that are the most distant from the threshold condition (e.g., upper left part of Fig. 3a) where it is more easy to have P high than low? Then, the n-dimensional surrogate will be

update by favoring these far-from threshold condition regions leading to a poor behavior (due to its global character) in those regions close to the threshold conditions (e.g., lower right region in Fig. 3). This is then reflected in the decreased quality of the behavioral parameters pdfs as shown in Fig. 4. Or maybe, I am just speculating too much here. It could be of interest to see how the n-dimensional surrogates evolve as a function of P, for example after some updates are conducted to see if there is this tendency or not.

> We are not quite sure we fully understand what the reviewer means here. The P-value states the minimum fraction of neighbours that has to be behavioral, and is hence in our view an exceedance number. We do, however, agree with the reviewer that the higher-P cases (i.e. requiring more neighbors to become stage-1 accepted) leads to a sampling that poorly samples the boundary regions (e.g. around the reg line in fig. 2). This is, as the reviewer also points out, clear from the results in figure 4, where the $P = 0.75$ case does not sample the margins of the histogram particularly well. However, we do not find any intuitive and easily understandable way of showing how the surrogate evolves, other than the clear results in Figure 4. Hence, based on this comment, no changes will be applied to the manuscript. However, if the reviewer has a clear suggestion we happy to learn about it!

Comment 5. Since a surrogate model is used to mimic also the full model response (see Sec. 2.2) I would suggest to refer to this as 'full-model-surrogate' in order to mark the distinction with the surrogate model build in the n-dimensional space.

> We see the reviewer's point, however, we rather like to avoid confusion by not naming the GPE-surrogate-model used as our virtual truth a surrogate. We will add this information to Section 2.2 and hope it makes the nomenclature clearer (lines 73-75):
>

[revised manuscript text omitted]